# Insecticidal Activity of 25 Essential Oils on the Stored Product Pest, *Sitophilus granarius*

**DOI:** 10.3390/foods10020200

**Published:** 2021-01-20

**Authors:** Sébastien Demeter, Olivier Lebbe, Florence Hecq, Stamatios C. Nicolis, Tierry Kenne Kemene, Henri Martin, Marie-Laure Fauconnier, Thierry Hance

**Affiliations:** 1Biodiversity Research Center, Earth and Life Institute, Université Catholique de Louvain, 4-5 Place Croix du Sud, 1348 Louvain-la-Neuve, Belgium; olivier.lebbe@uclouvain.be (O.L.); florence.hecq@gmail.com (F.H.); thierry.hance@uclouvain.be (T.H.); 2Interdisciplinary Center for Nonlinear Phenomena and Complex System, Université Libre de Bruxelles, Campus Plaine, CP 231 bd du Triomphe, 1050 Brussels, Belgium; stamatios.nicolis@ulb.be; 3Laboratory of Chemistry of Natural Molecules, Gembloux Agro-Bio Tech, Université de Liège, 2 Passage des Déportés, 5030 Gembloux, Belgium; kenne@gmx.com (T.K.K.); henri.martin@uliege.be (H.M.); marie-laure.fauconnier@uliege.be (M.-L.F.)

**Keywords:** essential oil, insecticide, eco-friendly, stored product pest, *Sitophilus granarius*, *Allium sativum*, *Gaultheria procumbens*, *Mentha arvensis*, *Eucalyptus dives*

## Abstract

The granary weevil *Sitophilus granarius* is a stored product pest found worldwide. Environmental damages, human health issues and the emergence of resistance are driving scientists to seeks alternatives to synthetic insecticides for its control. With low mammal toxicity and low persistence, essential oils are more and more being considered a potential alternative. In this study, we compare the toxicity of 25 essential oils, representing a large array of chemical compositions, on adult granary weevils. Bioassays indicated that *Allium sativum* was the most toxic essential oil, with the lowest calculated lethal concentration 90 (LC90) both after 24 h and 7 days. *Gaultheria procumbens*, *Mentha arvensis* and *Eucalyptus dives* oils appeared to have a good potential in terms of toxicity/cost ratio for further development of a plant-derived biocide. Low influence of exposure time was observed for most of essential oils. The methodology developed here offers the possibility to test a large array of essential oils in the same experimental bioassay and in a standardized way. It is a first step to the development of new biocide for alternative management strategies of stored product pests.

## 1. Introduction

Loss of food during storage by pest infestation is a major problem in our societies in both developed and developing countries, causing significant financial losses [1,2,3,4]. Stored cereals are, indeed, a source of food for many insects, mites and fungi which degrade the product quality and can cause from 9 to 20% of net losses [5]. Around 1660 insect species worldwide are known to affect the quality of stored food products [6]. Despite this worrying situation, few research funds are allocated to offset these losses [7].

Since 1960, stored product pests have been mainly controlled by synthetic contact pesticides [8,9]. The utilization of those pesticides is being criticized more and more. Appearance of resistance in addition to the increased risks of residues dangerous to the environment and human health have led to an increasingly restricted use of those compounds [9,10]. These environmental concerns and demand for food safety have underlined the need for alternative research [10,11]. In the last decades, plant essential oils have been reported to be a potential alternative for many applications such as anti-microbial, antifungal or herbicide uses [12]. More particularly, essential oils also have interesting properties to replace synthetic insecticides [13,14]. Isman and Grieneisen [15] showed that from 1980 to 2012 the proportion of papers on botanicals among all papers on insecticides raised from 1.43% to 21.38%. Increasing interest in essential oils as an alternative to synthetic pesticides comes from their characteristics [16]. Due to their high volatility, temperature and UV light degradation sensitivity, essential oils are less persistent in the environment than traditional pesticides [17]. In addition, most essential oils have low mammalian toxicity in comparison with synthetic insecticides and are considered as eco-friendly [18]. For instance, Stroh et al. [19] showed that eugenol was 1500 times less toxic than pyrethrum and 15,000 times less toxic than the organophosphate azinphosmethyl for juvenile rainbow trout based on 96 h-LC_50_ values.

In temperate regions, the granary weevil is considered as one of the major pests of stored grain [9,20,21,22]. Many authors [23,24,25,26,27,28,29] have investigated the use of essential oils as alternative insecticides against *S. granarius.* Yildirim et al. [30] demonstrated the high fumigation toxicity of *Satureja hortensis* among eleven essential oils from Lamiaceae family on *S. granarius*. Others have highlighted the contact toxicity by topical application of essential oils, such as Conti et al. [28] with *Hyptis* genera plants. A few less have worked on treated grains taking into account contact, fumigation and ingestion intoxication paths together [31]. The repellency potential of essential oils was also analysed [32,33] for *S. granarius*. In addition, essential oils were reported as a good food deterrent, as in the case of *H. spicigera* essential oils against *Sitophilus zeamais,* preventing grain degradation [34].

Nevertheless, few actual applications have emerged for the protection of stored foodstuffs and we still lack a systematic screening of potently active oils under conditions mimicking storage reality and with a standardized strain of insects. The aim of our study was precisely to test and rank 25 essential oils commonly used and available on the market against *S. granarius.* Special care has been taken for the selection of essential oil based on a large array of chemical composition (different major compounds or groups of major compounds, Table 1). In order to remain under realistic conditions for industrial large-scale application, data as price, availability on the market or health implications has been taken into account in our discussion. To allow comparison, a standardized strain of *S. granarius* was used for all the test performed under the same experimental conditions. Determination of essential oils toxicity has been done by treating the wheat grains directly, considering that the presence of wheat may influence results [35] and mostly because, in practice, it is the grain itself that will be treated in storage facilities.

## 2. Materials and Methods

### 2.1. Biological Material

The granary weevil, *S. granarius*, was collected in Belgium from infested wheat grain stock in 2016. They were reared at the Biodiversity Section of the Earth and Life Institute, under controlled conditions in a climatic chamber (28 °C ± 1, 75 ± 5% RH, in the dark) on organic wheat (*Triticum aestivum*).

### 2.2. Selected Essential Oils (EO) and Their Composition

EOs were selected based on their availability on the market and their composition. Selected essential oils have all a distinctive major compound or a combination of major compounds to make sure to test a large range of composition.

Essential oils have been mainly obtained from Pranarom S.A. (7822—Ghislenghien, Belgium) as well as their composition. Only *Ocimum sanctum* essential oil has been purchased from “Herb and tradition” S.A company (CP59560—Comines, France) and was analyzed by GC-MS. List of the essential oils tested and their composition is indicated in Table 1. The GC-MS used for EOs characterization was a Hewlett Packard system (HP Inc., Palo Alto, CA, USA) in splitless injection mode system, with a HP INNOWAX column of 60 m, 0.25 mm of diameter and 0.5 µm of film thickness. The initial temperature of 50 °C was maintained for 6 min before a progressive warming of 2 °C per minute up to 250 °C. Once the temperature peak of 250 °C was reached, it has been maintained for 20 min. The injector and interface temperature were 250 °C whereas that of the source was 230 °C. The gas vector was helium at a pressure of 23 psi and the total ion chromatogram was recorded by using an electron-impact source at 70 eV of ion kinetic energy. The compound identification was made by comparison of the spectra to National Institute of Standards and Technology (NIST, Gaithersburg, MD, USA) spectral library and pure commercial standards injection in the same chromatographic conditions.

### 2.3. Toxicity Test in Treated Grain

To be as close as possible to realistic application conditions, we have chosen to treat the grains directly with a standardized quantity of oils. A determined quantity of insects of the same age group was then directly placed on the grains. Consequently, the observed mortality is a result of contact with the treated grain, attempts at nutrition and a fumigation effect.

Toxicity tests were performed in 15 mL plastic Falcon tubes containing 8 g of treated wheat. One mL of essential oil diluted in acetone at concentrations of respectively 1; 2; 3; 4 and 5% (*v/v*) were applied on the wheat except for *Gaultheria procumbens* for which concentrations of 5; 3; 2; 1 and 0.75% were used. Moreover, because of its efficiency, the same tests of mortality have been realized for *Allium sativum* at lower concentrations of 0.75%; 0.5%, 0.25% and 0.125%.

After EO application, samples were mixed by a vortex for 1 min to homogenize the treatment. The control treatment consisted of five Falcons with 8 g of wheat treated with 1 mL of acetone only. Treated wheat dried for 15 min under hood to eliminate the acetone. Then, twenty insects per falcon were added to the wheat and Falcon were closed by a tulle to allow air circulation. Tubes were placed under controlled condition (28 °C ± 1 °C; 75 ± 5% RH). Temperature and humidity were chosen as the optimum for *S. granarius* [36] and to be representative of the conditions at the harvest period. Five repetitions were performed for each concentration.

The mortality was recorded after 24 h and 7 days of exposure. Light is repellent to *S. granarius* [37]. This particularity was used to identify dead individuals by placing a cold lamp of 100 watt in front of eyes of insects for 5 s. Individuals unable to move were considered dead.

### 2.4. Data Analysis

In the control treatment, in one case the average mortality reached 5 percent and consequently, the Abbott formula [38] has been used to correct mortality.

The relationship between the mortality rate and the concentrations of the different oils tested was fitted with a Hill function using Scipy module of Python v.3.8.2 (Beaverton, OR, USA). This allowed us to estimate the LC50 (lethal concentration that produces 50% of mortality) and LC90 (lethal concentration that produces 90% of mortality). The Hill function is frequently used in different disciplines, from biochemistry and cellular biology to Physics [39] with the following Equation (1):(1)$$M=\frac{C^{n}}{C^{n}+K^{n}}$$
where M is the mortality proportion; C is the concentration of oil used; K a threshold concentration value beyond which the mortality exceeds 50% (which corresponds to the LC50) and n a cooperativity exponent. A value of n that is larger than 1 signals the presence of cooperative processes between the concentration level and the propagation of the mortality inside the population. In order to calculate the resulting LC for an arbitrary proportion of the population by rearranging the previous Equation (1):(2)$$C_{x}=\;{\left(\frac{M_{x}}{1-M_{x}}\right)}^{1/n}K$$
which as in Equation (3) gives for the *LC*90
(3)$$C_{90}=LC90=\;9^{1/n}K$$

LC90 has been used to compare essential oils’ toxicity. Toxicities are considered significantly different if its standard deviation does not overlap.

## 3. Results

### Mortality Analyses

Mortality levels clearly varied among oils. When tested at the highest concentration of 5%, nine out of 25 essential oils provoked a mortality of less than 60% of the individuals after 24 h (ranging from 0 to 59%). We considered that this threshold must be exceeded to give sufficient efficiency in practice. In consequence, for these oils lower concentrations were not further tested. Looking at the results, it appeared that EOs listed in Table 2 are not effective at this concentration on *S. granarius.*

For the 16 remaining EOs, a positive relation was observed between mortality and concentration. Most of them showed a zero or almost zero mortality at a concentration of 1% except *A. sativum* which still provoked 75% of mortality after 24 h at that concentration and represents therefore the most toxic oil tested. Among the remaining oil, *G. procumbens, O. sanctum* and *Eucalyptus* dives reached respectively 81%, 68% and 51% of mortality (24 h) for a 2% concentration (Table 3).

Calculation of mortality curves was realized for 24 h and 7 days treatment (Figure 1). Table 4 indicates the LC90 after 24 h and 7 days for these 16 essential oils tested.

For most of EOs tested, time of exposure did not have a significant effect on percentage of mortality, indicating that a knock down effect is rapidly observed (Table 4). However, this observation does not hold for three EOs after 24 h and 7 days, *Thymus vulgaris* CT geraniol, *Myristica fragrans* and *O. sanctum,* indicating a cumulative toxic effect probably linked to physiological or neurological disorders.

With a LC90 of 1.04% after 7 days of exposure, *A. sativum* is the most toxic essential oil tested. It is followed by *G. procumbens* and *O. sanctum* that showed similar results with LC90 of 2.10 and 2.11% (7 days). The third position in the list of the most toxic essential oils is shared by *Mentha arvensis, T. vulgaris* CT geraniol and *E. dives* which present respectively a LC90 of 3.08; 3.08 and 3.11% after 7 days of exposure.

## 4. Discussion

### 4.1. Insecticidal Potential

This study compares the toxicity of 25 essential oils on the granary weevil. Sixteen of these were found to have an interesting insecticidal activity on *S. granarius*. Our results show that *A. sativum*, *G. procumbens*, *O. sanctum*, *M. arvensis, T. vulgaris* (geraniol) and *E. dives* present a potential to control *S. granarius* population directly in the grain.

Garlic essential oil has been identified as the most toxic oil with a LC90 two to four times lower than other EOs, probably because of its content in sulfur compounds. Its toxicity on other insect pests of stored products like *Tenebrio molitor* [40], *Sitotroga cerealella* [41], *Tribolium castaneum* and *Sitophilus zeamais* [42,43] has already been described. The efficiency of garlic essential oils and his constituents may vary with the target species, the stage of life and the exposure mode (fumigation or contact). For example, Ho et al. [42] calculated a KD50 (knock down) of 1.32 mg/cm^2^ and 7.65 mg/cm^2^ of garlic essential oil against *T. castaneum* and *S. zeamais* respectively. In addition, Plata-Rueda et al. [40] have identified diallyl disulfide as the most toxic compounds present in the garlic essential oil explaining its efficiency on *Tenebrio molitor.* Contact and fumigation toxicities of diallyl trisulfide has been highlighted by Huang et al. [43] on *T. castaneum* and *S. zeamais*. Contrary to most other essential oils, these molecules are not present in the garlic clove itself, but arise from the conversion of thiosulfinates (water-soluble) to sulfides (oil-soluble) during the hydrodistillation process [44]. In short, the main sulfur compounds in the whole garlic clove are cysteine sulfoxides like allylcysteine sulfoxide (alliine) and methylcysteine sulfoxide (methiine) which are located in the clove mesophyll storage cells. After crushing the clove, those compounds come in contact with the enzyme *alliinase* that is normally localized in the vascular bundle sheath cells. The vast majority of cysteine sulfoxides are then converted in sulfenic acids which self-condense to thiosulfinates like allicin which is the most abundant compound (60–90% of total thiosulfinate). Allicin is quite unstable depending on the medium and temperature. Upon hydrodistillation, thiosulfinates are transformed into diallyl trisulfide, diallyl disulfide and allyl methyl trisulfide as major products [44].

Essential oils toxicity of *M. avensis* [45], *G. procumbens* [46] and *E. dives* [47] as well as geraniol (main compound of *T. vulgaris* essential oil) [48] was also been highlighted for their activities against various stored product pests. In addition, Yazdgerdian et al. [46] identified *G. procumbens* as the most toxic oil, both by fumigation (6.8 µL/L air) and contact on treated wheat (0.235µL/g), among five essential oils tested on *S. oryzae*. These results confirm the toxicity of *G. procumbens* observed in our study. However, although many studies highlighted toxicities of essential oils, lack of a common protocol or of major compounds description often prevent from reliable and univocal comparison. For example, in the study of Teke et al. [49] the fennel essential oils applied on *S. granarius* contains 71.64% of estragol, which closely resembles the composition of the basil oil in our study (73.43% estragol). However, in their case they realized topical application without grain presence which is quite different that in our case.

At the opposite, Zohry et al. [50] tested toxicity of 10 essential oils on *S. granarius* by exposure to treated wheat in a protocol close to that of this study. Garlic oil was identified as the most toxic one with a concentration of oil per grams of grain similar to ours. However, no precision on composition of EOs are available in their publication, which do not allow a deeper comparison. Further studies on the evaluation of the industrial potential of essential oils need to be based on a common protocol taking into account the influence of the media [35] and a full description of the composition of essential oils.

Despite numerous studies on the toxicity of essential oil on stored product pests, little data is available on the mechanism of action of the insecticidal effect of these essential oils as a mixture of molecules. However, some studies highlight some mechanisms. For instance, Jankowska et al. [51] showed that menthol acts on octopamine receptors and trigger protein kinase A phosphorylation pathway on cockroach DUM neurons. Hong et al. [52] indicate a potential interference of methyl salicylate and eugenol with octopamynergic system as well. Action on octopamine receptors is an advantage in the elaboration of an insecticide due to absence of key role in vertebrates involving a relative security for human health. However, methyl salicylate is known to have a LD50 oral (rat) of 887 mg/kg indicating that it should have another mechanism of action on mammals. Therefore, the mere fact that octopamynergic system is targeted by an essential oil cannot guarantee safety for human health. β-caryophyllene was identified as an inhibitor of the activities of acetylcholine esterase, polyphenol oxidase and carboxylesterase on *Aphis gossypii* [53]. *α*-phellandrene is believed to have a neurotoxic effect on *Lucinia cuprina* [54]. Diallyl disulfide is known to impact digestion of *Ephestia kuehniella* by decreasing activity of digestive enzymes [55]. Diallyl trisulfide, another major compound of garlic EO, has been recently described as a regulator of the expression of the chitin synthase A gene which generates alteration of the morphology and inhibition of the oviposition of *Sitotroga cerealella* [56]. Finally, essential oils are complex mixture of molecules, possibly interacting and entering in synergy for their mechanism of action. Therefore, it is important to analyse their impact on insect as a whole. For example, a recent study shows that *M. arvensis* EO is associated with a systemic mode of action on *S. granarius* since it is capable of altering the nervous and muscular systems, cellular respiration processes and the cuticle, the first protective barrier of insects [57].

### 4.2. Human Health Risk

Toxicity on the target pests is a first step for any kind of new pesticide elaboration. However, in the perspective of a potential utilization of essential oils in an industrial context, it is also essential to focus on some other aspects, such as the price, the wheat deterioration or the mammal toxicity to determine their actual industrial potential. Concerning mammal toxicity, the WHO classification ranked compounds from “extremely hazardous” to “unlikely to present acute hazard” based on the concentration in mg/kg that provoke 50% of mortality in rat (WHO, 2009). Concerning *A. sativum,* diallyl trisulfide is ranked as “unlikely to present acute hazard” while diallyl disulfide is considered as moderately hazardous with an oral LD50 (rat) of 260 mg/kg. Even if this toxicity is two to four times lower than deltamethrin currently used in granaries, it remains to be carefully considered in the case of a conception of healthy and ecofriendly alternatives to insecticide.

*Gaultheria procumbens* which showed the second highest acute toxicity to *S. granarius* is constituted at 99% of methyl salicylate, a molecule classified as moderately hazardous for human health. Because of this specific composition, this essential oil should be use in association to avoid a rapid development of resistance. Further analyses have also to be done on the persistence of methyl salicylate, on its environmental and mammal toxicity to estimate the potential of this EO as a stand-alone or mix product. Two molecules of *O. sanctum* (eugenol and methyl eugenol) as well are classified as moderately hazardous to mammals and need to be considered with the same caution.

For the three other oils identified, major compounds are all classified as “slightly hazardous” to “unlikely to present acute hazard” and their use should not be a problem to treat food product.

### 4.3. Prices

If we considered prices (Table 5), essential oils are quite expensive, particularly garlic oil probably because its low availability and its use mainly as an aroma in food industry. Moreover, sulfides are also well known for their unpleasant odor complicating its practical application. These two points explained its low practical applications. *O. sanctum* also seems too expensive to be used at an industrial scale.

*Gaultheria procumbens, M. arvensis* and *E. dives* are among the less costly essential oils on the market. Moreover, these three oils are easily available on the market. Based on our results, their toxicity and their price, these three essential oils could represent good opportunity to develop a botanical insecticide to control insect pest in stored product. We did not obtain a commercial price for *T. vulgaris* CT geraniol at an industrial scale.

### 4.4. Duration of Exposure

Only three essential oils (*M. fragrans, O. sanctum* and *T. vulgaris* CT geraniol) showed an increase in mortality 7 days after the treatment (Figure 1). This could be the consequence of a cumulative contamination during all the period, including by feeding. It is also possible that physiological disorders took times and was linked to an arrest of feeding and water losses.

For the other essential oils, little differences of mortality were observed after 1 and 7 days of exposure. Several hypotheses could explain that observation. First as mortality arise soon after the insect introduction, we may expect a strong selection effect on susceptible individuals, leaving alive after one day only more resistant individuals. Secondly, the absorption of essential oils by the grains (by fumigation or contact) could reduce the biodisponibility of the active compounds and thus the lack of efficiency on long terms period. Indeed, Lee et al. [35] put into light that fumigation toxicity of certain essential oils is three to nine times lesser in presence of wheat due to the absorption phenomena.

Thirdly, our experiment has been conducted at 28 °C. The evaporation rate of essential oils is rapid at this temperature and a substantial part of the essential oil may have vanished after 24 h. Heydarzade et al. [58] highlighted the low persistence of essential oils of *Teucrium polium* and *Foeniculum vulgare.* Treated filter paper induced a 99% mortality at time zero and 0% 30 h after application on *Callosobruchus maculatus* adults. This downgrade of activity is supposed to be caused by high volatility and/or quick degradation of active compounds.

Studies must be carried out on the combined influence of evaporation and absorption by grains of essential oils in order to demonstrate their toxicity persistence over time. In further studies, it is a priority to include GC-MS analyses of treated wheat that allowed scientists to determine the behavior of essential oils and its remanence at the surface and inside the treated wheat until the end of experiment. This factor is essential to control insect pests that lay eggs into the grain, which causes a delay between treatment and the potential contact with the insecticide product by emerging individuals.

Finally, we cannot exclude that the low difference between mortalities for both exposure times could be explained by the absence of accumulation of toxic compounds in the insect and its capacity to metabolize them. The few cases where a difference was identified between both exposure times could be explain by a more physiologic mode of action inducing drying or no feeding effect which induces slower death pattern.

## 5. Perspectives

Moreover, to precise if these essential oils could be a viable alternative to pesticide in an industrial point of view, further studies has to be conduct on the comparison of their efficiency with the one of actual synthetic insecticides and/or natural substances well known for their insecticidal properties in a protocol mimicking the actual mode of treatment. To answer eventually the question: “Are these essential oils actually a good alternative to the current standards”, future studies should include a positive control with a treatment protocol based on pulverization.

Experiments should also be carried out at a larger scale, such as experimental granaries, with the purpose of estimating the quantity of oil per ton of wheat needed and thus the practical applicability of these treatments. Indeed, under mass storage conditions, the application of essential oils during the grain filing process in the silo is based on nano-drop pulverization which could greatly increase the evaporation of the product. Moreover, the formulation of the essential oil is also of tremendous importance as discussed by Maes et al. [59]. In our cases, dilutions were made using acetone which is also quite different from actual industrial application. These points should be further analyzed in details.

## 6. Conclusions

Considering insecticidal effects, prices, availability and mammal toxicity of essential oils tested, *M. arvensis, E. dives* and *G. procumbens* can be considered as good potential alternatives to the synthetic pesticides presently used to control grain weevils. As essential oils are products of very variable composition, studies must be performed to clearly identify the compound(s) responsible of the insecticidal toxicity of these three essential oils to avoid variable responses to future treatments. More investigations need to be done on the mechanism of action of these oils, including the role of minor components, both on insects and mammals, to secure their industrial use.

## Figures and Tables

**Figure 1 foods-10-00200-f001:**
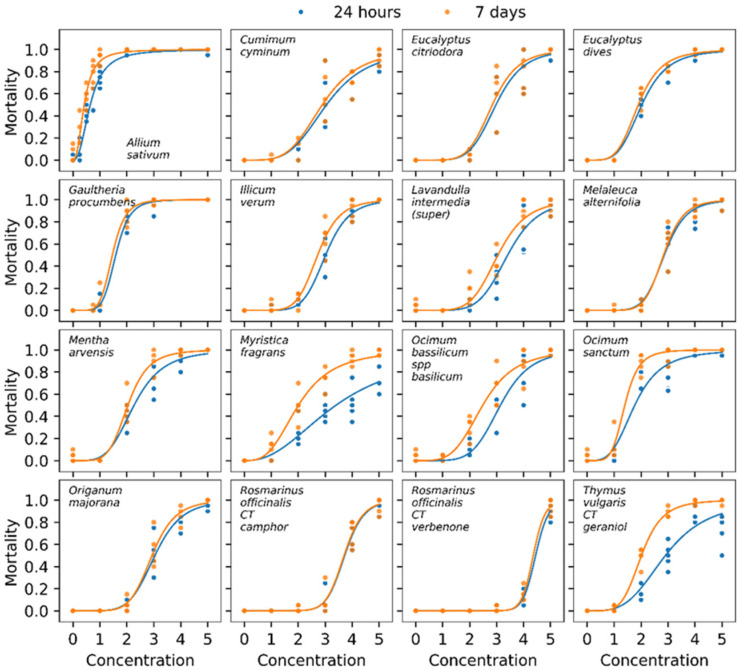
Mortality curves of EOs tested on *S. granarius* 24 h after treatment (blue) and 7 days after treatment (orange).

**Table 1 foods-10-00200-t001:** List of the essential oils tested and their composition for compounds (main compounds representing more than 10% of the total composition on peak area basis).

Essential Oils	Major Compounds (>10%)	Essential Oils	Major Compounds (>10%)
*Abies sibirica* Ledeb.	Bornyl acetate (20.41%), camphene (19.51%), limonene (18.04%), α-pinene (15.71%)	*Melaleuca alternifolia* (Maiden & Betche) Cheel	Terpinene-4-ol (40.14%), γ-terpinene (18.75%)
*Allium sativum* L.	Diallyl disulfide (36.60%), diallyl trisulfide (32.33%)	*Mentha arvensis* L.	Menthol (73.72%)
*Cinnamomum camphora (*L.*) J. Presl ct cineole*	1,8-cineole (53.11%), sabinene (14.50%)	*Myristica fragrans* Houtt.	α-Pinene + α-thujene (21.78%), sabinene (17.91%), β-pinene (14.68%)
*Citrus limon (*L.*)* Burm. F.	Limonene (68.13%), β-pinene (12.04%)	*Myrtus communis* L.	α-Pinene (54.71%), 1,8-cineole (24.31%)
*Copaifera officinalis* L.	β-Caryophyllene (64.25%)	*Ocimum basilicum* L.	Estragol (73.43%), linalool (18.85%)
*Cuminum cyminum* L.	Cuminaldehyde (28.11%), γ-terpinene (20.88%), *p*-cymene (18.26%), β-pinene (14.18%)	*Ocimum sanctum* L.	Eugenol (33.7%), β-caryophyllene (21.8%), methyleugenol (20.5%)
*Eucalyptus citriodora* Hook	Citronellal (71.09%)	*Origanum majorana* L.	Terpinen-4-ol (21.67%), *cis*-thujanol (15.69%), γ-terpinene (14.14%)
*Eucalyptus dives* Schauer	Piperitone (47.87%), α-phellandrene (23.33%)	*Rosmarinus officinalis* L. CT camphor	α-pinene (24.62%), 1,8-cineole (16.43%), camphor (16%), camphene (10.90%)
*Eucalyptus globulus* Labill.	1,8-Cineole (66.10%)	*Rosmarinus officinalis* L. CT verbenone	α-Pinene + α-thujene (31.84%), camphor (10.65%)
*Gaultheria procumbens* L.	Methyl salicylate (99%)	*Thymus vulgaris* L. CT geraniol	Geraniol (58.25%), geranyl acetate (14.03%)
*Illicium verum* Hook. F.	*trans*-Anethole (77.71%)	*Vetiveria zizanioides* (L.) Stapf	Khuenic acid (10.48%)
*Lavandula hybrida super*	Linalool (33.90%), linalyl acetate (33.20%)	*Zingiber officinale* Roscoe	α-Zingiberene (19.87%), β-sesquiphellandrene (14.64%), camphene (12.18%)
*Matricaria recutita* (L.) Rauschert	*E*-(*trans*)-β-farnesene (41.17%)	*-*	-

**Table 2 foods-10-00200-t002:** Essential oils tested at a concentration of 5% for which the mortality was not satisfactory.

Essential Oil	Major Compounds	Mortality (24 h)	Control Mortality (24 h)
*Cinnamomum camphora CT cinéole*	1,8 Cineole (53.11%), sabinene (14.50%)	59% ± 10.2	0%
*Zingiber officinale*	α-Zingiberene (19.87%), β-sesquiphellandrene (14.64%), camphene (12.18%)	45% ± −5.5	0%
*Eucalyptus globulus*	1,8-Cineole (66.10%)	33% ± 9.1	0%
*Abies sibirica*	Bornyl acetate (20.41%), camphene (19.51%), limonene (18.04%), α-pinene (15.71%)	9% ± 10.2	0%
*Matricaria recutita*	*E*-(*trans*)-β-Farnesene (41.17%)	7% ± 1.1	0%
*Copaifera officinalis*	β-Caryophyllene (64.25%)	5% ± 6.3	0%
*Vetiveria zizanoides*	Khuenic Acid (10.48%)	2% ± 0.5	0%
*Citrus limon*	Limonene (68.13%), β-pinene (12.04%)	0%	0%
*Myrtus communis*	α-Pinene (54.71%), 1,8 cineole (24.31%)	0%	1% ± 0.45

**Table 3 foods-10-00200-t003:** Summary of mortality percentages after 24 h hours of exposure for the concentrations tested (*n* = 5).

Essential Oils	5%	4%	3%	2%	1%	Control
*A. sativum*	99 ± 2.2%	100%	100%	98 ± 2.8%	75 ± 7.9%	1 ± 0.45%
*C. cyminum*	90 ± 7.1%	68 ± 12.5%	55 ± 25%	11 ± 6.5%	0%	0%
*E. citriodora*	96 ± 5.5%	79 ± 16.3%	56 ± 18.5%	3 ± 4.5%	0%	0%
*E. dives*	100%	96 ± 4.2%	80 ± 6.1%	51 ± 7.4%	0%	0%
*G. procumbens*	100%	-	96 ± 4.2%	81 ± 6.5%	5 ± 6.1%	0%
*I. verum*	100%	87 ± 5.7%	50 ± 12.7%	7 ± 4.5%	0%	0%
*L. intermedia super*	88.89 ± 5.5%	80.81 ± 16.7%	30.3 ± 12.9%	5 ± 5%	0%	0%
*M. alternifolia*	98 ± 4.5%	86.87 ± 9.2%	61 ± 15.6%	3 ± 4.5%	2 ± 4%	0%
*M. arvensis*	100%	93 ± 8.4%	73 ± 13.0%	41 ± 10.8%	0%	0%
*M. fragrans*	75 ± 17.3%	52 ± 14.8%	46 ± 9.6%	25 ± 11.7%	0%	0%
*O. basilicum spp basilicum*	97 ± 2.7%	80 ± 19.7%	41 ± 14.7%	12 ± 5.7%	0%	0%
*O. sanctum*	99 ± 2.3%	98 ± 2.8%	75.75 ± 11.5%	68 ± 6.7%	6 ± 4.2%	0%
*O. majorana*	97 ± 4.5%	81 ± 6.5%	50 ± 16.6%	4 ± 4.2%	0%	0%
*R. officinalis CT camphor*	93 ± 7.6%	69 ± 10.8%	8 ± 9.7%	0%	0%	0%
*R. officinalis CT verbenone*	90 ± 7.9%	12 ± 5.7%	2 ± 2.7%	0%	0%	0%
*T. vulgaris CT geraniol*	74 ± 14.7%	89 ± 8.2%	50 ± 11.2%	20 ± 7.1%	0%	0%

**Table 4 foods-10-00200-t004:** Summary of mortality data presented at the Figure 1 for the 16 essential oils tested. Lethal concentrations are expressed in percent.

Essential Oil	Exposure Time	LC50	LC90	*R* ^2^	*n*
*Allium sativum*	24 h	0.64 ± 0.02	1.43 ≤ 1.58 ≤ 1.75	0.983	2.4 ± 0.19
	7 days	0.42 ± 0.02	0.93 ≤ 1.04 ≤ 1.17	0.976	2.4 ±0.20
*Cumimum cyminum*	24 h	3.05 ± 0.12	4.72 ≤ 5.27 ≤ 6.02	0.942	4.0 ± 0.59
	7 days	2.89 ± 0.10	4.42 ≤ 4.88 ≤ 5.50	0.952	4.2 ± 0.57
*Eucalyptus citriodora*	24 h	2.98 ± 0.08	4.02 ≤ 4.34 ≤ 4.77	0.956	5.8 ± 0.88
	7 days	2.84 ± 0.09	3.76 ≤ 4.11 ≤ 4.58	0.945	6 ± 1.03
*Eucalyptus dives*	24 h	2.03 ± 0.04	3.21 ≤ 3.4 ≤ 3.61	0.991	4.3 ± 0.33
	7 days	1.90 ± 0.038	2.94 ≤ 3.11 ≤ 3.32	0.991	4.4 ± 0.37
*Gaultheria procumbens*	24 h	1.59 ± 0.04	2.15 ≤ 2.26 ≤ 2.4	0.993	6.2 ± 0.55
	7 days	1.46 ± 0.04	1.99 ≤ 2.10 ≤ 2.23	0.99	6.1 ± 0.48
*Illicum verum*	24 h	3.02 ± 0.04	3.97 ≤ 4.14 ≤ 4.35	0.986	6.9 ± 0.68
	7 days	2.72 ± 0.05	3.58 ≤ 3.78 ≤ 4.01	0.98	6.7 ± 0.74
*Lavandulla intermedia (super)*	24 h	3.41 ± 0.08	4.57 ≤ 4.89 ≤ 5.29	0.958	6.1 ± 0.81
	7 days	3.05 ± 0.08	4.16 ≤ 4.48 ≤ 4.90	0.96	5.7 ± 0.82
*Melaleuca alternifolia*	24 h	2.86 ± 0.06	3.67 ≤ 3.89 ≤ 4.16	0.976	7.2 ± 0.98
	7 days	2.84 ± 0.05	3.56 ≤ 3.76 ≤ 4.02	0.979	7.8 ± 1.11
*Mentha arvensis*	24 h	2.27 ± 0.06	3.55 ≤ 3.83 ≤ 4.16	0.98	4.2 ± 0.42
	7 days	2.04 ± 0.05	2.86 ≤ 3.08 ≤ 3.36	0.98	5.3 ± 0.73
*Myristica fragrans*	24 h	3.40 ± 0.17	7.31 ≤ 8.68 ≤ 10.72	0.946	2.3 ± 0.35
	7 days	2.01 ± 0.07	3.69 ≤ 4.09 ≤ 4.59	0.975	3.1 ± 0.31
*Ocimum bassilicum spp basilicum*	24 h	3.14 ± 0.08	4.30 ≤ 4.63 ≤ 5.05	0.961	5.7 ± 0.78
	7 days	2.49 ± 0.08	3.87 ≤ 4.24 ≤ 4.73	0.964	4.1 ± 0.50
*Ocimum sanctum*	24 h	1.77 ± 0.07	2.94 ≤ 3.26 ≤ 3.66	0.973	3.6 ± 0.40
	7 days	1.40 ± 0.05	1.96 ≤ 2.11 ≤ 2.27	0.981	5.4 ± 0.51
*Origanum majorana*	24 h	3.04 ± 0.06	4.13 ≤ 4.36 ≤ 4.65	0.978	6.1 ± 0.67
	7 days	2.94 ± 0.05	3.85 ≤ 4.06 ≤ 4.32	0.98	6.8 ± 0.81
*Rosmarinus officinalis* CT camphor	24 h	3.72 ± 0.04	4.43 ≤ 4.58 ≤ 4.75	0.981	10.6 ± 1.16
	7 days	3.70 ± 0.05	4.39 ≤ 4.54 ≤ 4.73	0.978	10.7 ± 1.26
*Rosmarinus officinalis* CT verbenone	24 h	4.45 ± 0.03	4.93 ≤ 5.00 ≤ 5.07	0.99	18.8 ± 1.14
	7 days	4.36 ± 0.03	4.80 ≤ 4.86 ≤ 4.94	0.992	20.1 ± 1.41
*Thymus vulgaris* CT geraniol	24 h	2.90 ± 0.11	4.75 ≤ 5.33 ≤ 6.10	0.948	3.6 ± 0.5
	7 days	2.02 ± 0.03	2.95 ≤ 3.08 ≤ 3.23	0.994	5.2 ± 0.38

**Table 5 foods-10-00200-t005:** Price of the most lethal oils tested and the mammal’s toxicity of their major compounds.

Essential Oil	Price ($/kg)	Major Compounds	DL50 Oral Rat Toxicity (Mg/Kg)	WHO Classification
*A. sativum*	130–250	Diallyl disulfide (36.6%)	260 *	II
		Diallyl trisulfide (32.33%)	5800 *	U
*G. procumbens*	55	Methyl salicylate (99%)	887 **	II
*E. dives*	34	Piperitone (47.87%)	3350 ***	III
		α-phellandrene (23.33%)	5700 *	U
*M. arvensis*	22	Menthol (73.72%)	3300 **	III
*O. sanctum*	200	Eugenol (33.7%)	1930 **	II
		β-caryophyllene (21.8%)	>5000 ****	U
		Methyl eugenol (20.5%)	810 *****	II
*T. vulgaris* CT geraniol	-	Geraniol (58.25%)	3600 **	III
		Geranyl acetate (14.03%)	6330 *	U

Data obtained from safety data sheet from: * Cayman (Ann Arbor, MI, USA); ** Fisher Science education (Rochester, NY, USA); *** Echemi (Qingdao, China); **** Carl Roth (D-76185 Karlsruhe, Germany); ***** CDH Fine Chemicals (New Delhi, India). Prices have been obtained from Ultra Internationnal B.V. (Spijkenisse, The Netherlands). WHO Classification: II: Moderately hazardous; III: Slightly hazardous; U: Unlikely to present acute hazard.

## Data Availability

The data presented in this study are openly available in 10.6084/m9.figshare.13603526.

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
