# Peer review of "Insecticidal Activity of 25 Essential Oils on the Stored Product Pest, Sitophilus granarius"

_foods, 2021, doi:10.3390/foods10020200_

Round 1
Reviewer 1 Report
L44. The reference to anti-cancer activity should be eliminated, since this is an evidence emerging from in vitro evaluations and not connected to the topic of the paper.
No control with a well known insecticide is provided, with a strong limitation on data evaluation. I would suggest the recourse to both a commercial insecticide commonly used against Sitophilus spp. and a natural substance well known for its insecticidal properties. This would put the results in a proper context, thus providing a more reliable answer to the following question: "is there any essential oil actually useful as an alternative to the current standards available?"
The correlation between bioactivity and phytochemical profile of essential oils is quite limited. The partial indication of the content of the oils is suitable for describing the chemotypes used, but insufficient to bring out complete links between composition and activity, especially when the synergies mentioned by the authors themselves may be present. The use of multivariate analysis on complete analytical data would allow a better description of trends based on groups of compounds and functional gropus and provide a more reliable grouping of the different oils.
L275 There is no reasoning or verification regarding the permanence of off-flavor or residues, thus limiting any discussion regarding toxicity and long time effects. A targeted GC-MS analysis of the grain at the end of the experimental period would be useful to verify concretely and quantitatively the actual presence of residues.
L150 it is not clear how the contribution of these compounds was determined in the absence of tests done with pure compounds. Also in this case, the use of an in-depth multivariate analysis could help, although some experiments on single pure compounds could also be very useful to minimize the risk of variable responses due to the use of often very changing mixtures such as essential oils. Such approach would also provide a new dimension to the manuscript, whose novelty is somehow limted.
L190: Was toxicity to these insects observed with similar values? More or less intense?
L192 Some papers cited report toxicity expressed per mg / cm3. Would it be possible to extrapolate the data obtained in these experiments in the same way, to facilitate comparison?
It might be useful to integrate the discussion with some recently published articles dedicated to the same topic, also to highlight the heterogeneity of the studies that prevents reliable and univocal comparisons.- Teke, M. A., & Mutlu, Ç. (2020). Insecticidal and behavioral effects of some plant essential oils against Sitophilus granarius L. and Tribolium castaneum (Herbst). Journal of Plant Diseases and Protection, 1-11.
Zohry, N. M., Ali, S. A., & Ibrahim, A. A. (2020). Toxicity of ten Native edible and essential plant oils against the Granary Weevil, Sitophilus granarius L.(Coleoptera: Curculionidae). Egyptian Academic Journal of Biological Sciences, F. Toxicology & Pest Control, 12(2), 219-227. - Shafaie, F., ARAMIdEH, S. H. A. H. R. A. M., VALIzAdEgAN, O. R. U. J., SAFARALIzAdEH, M. H., & Hosseini-Gharalari, A. (2019). Efficacy of Herbal Essential Oils against Cowpea Weevil, Callosobruchus maculatus Fabricus, and Wheat Weevil, Sitophilus granarius L. Oriental Journal of Chemistry, 35(3), 1174.
- Zohry, N. M., Ali, S. A., & Ibrahim, A. A. (2020). Repellency of Ten Edible and Essential Native Plant Oils to The Granary Weevil, Sitophilus granarius L.(Coleoptera: Curculionidae). Egyptian Academic Journal of Biological Sciences. A, Entomology, 13(4), 187-197.
Author Response
Dear Reviewer,
I would like to thank you for the time you devoted to proofreading our manuscript.
I will answer by taking your questions one by one :
Point 1 : L44. The reference to anti-cancer activity should be eliminated, since this is an evidence emerging from in vitro evaluations and not connected to the topic of the paper.
Response 1: Thank you for your suggestion; we removed it from the text and the references section.
Point 2: No control with a well known insecticide is provided, with a strong limitation on data evaluation. I would suggest the recourse to both a commercial insecticide commonly used against Sitophilus spp. and a natural substance well known for its insecticidal properties. This would put the results in a proper context, thus providing a more reliable answer to the following question: "is there any essential oil actually useful as an alternative to the current standards available?"
Thank you for this suggestion. However, our main objective was to compare essential oils between them for their activities as a first step on the way to find alternative to pesticide. The use of pesticide as a positive control will be useful in a second step, notably to establish the amount to apply and to compare prices of application with the most toxic essential oils identified. Moreover, the way of insecticide application (pulverization) is quite different as here we focus more on direct application on the grain. To compare actual insecticide to essential oils, we recommend doing it with a pulverization protocol to adjust the comparison to the final use. Toxicity of usual insecticides for store products protection is already well established and described in the literature (see Kljajić, P., & Perić, I. (2006). Susceptibility to contact insecticides of granary weevil Sitophilus granarius (L.)(Coleoptera: Curculionidae) originating from different locations in the former Yugoslavia. Journal of Stored Products Research, 42(2), 149-161.)
However, to take into account your proposal, we have added this text at the end of the first § of conclusions and perspective.
Considering insecticidal effects, prices, availability and mammal toxicity of essential oils tested, M. arvensis, E. dives and G. procumbens can be considered as good potential alternatives the synthetic pesticides presently used to control grain weevils. More investigations need to be done on the mechanism of action of these oils, including the role of minor components, both on insects and mammals to secure their industrial use. Moreover, to precise if these essential oils could be a viable alternative to pesticide in an industrial point of view, further studies has to be conduct on the comparison of their efficiency with the one of actual synthetic insecticide and/or natural substance well known for its insecticidal properties in a protocol mimicking the actual way of treatment. To answer eventually at the question: “ is those essential oils are actually a good alternative to the current standards”, future studies need to include a positive control with a protocol of treatment based on pulverization.
Point 3 : The correlation between bioactivity and phytochemical profile of essential oils is quite limited. The partial indication of the content of the oils is suitable for describing the chemotypes used, but insufficient to bring out complete links between composition and activity, especially when the synergies mentioned by the authors themselves may be present. The use of multivariate analysis on complete analytical data would allow a better description of trends based on groups of compounds and functional gropus and provide a more reliable grouping of the different oils.
Response 3: Within the framework of this study, we have voluntarily chosen to work with common commercial essential oils whose chemical composition is well known in order to come as close as possible to industrial use. Each essential oil comprises several dozen different constituents, which is why we have not presented detailed results of their compositions (especially as these are commercial essential oils). If it is your wish, we can add these data as additional information.
Your suggestion to use a multivariate analysis is interesting to try to correlate the chemical composition with the insecticidal effect, but in our case, given the large number of essential oils tested and their complex and very different composition (chosen voluntarily), this type of study could prove to be inconclusive.
On the other hand, it can never be excluded that a compound present in small proportion is very active or that synergies/antagonisms of action may occur.
We will keep your suggestion in mind for further studies on a smaller number of essential oils aiming to characterize the mode of action more finely. In this context, working with pure constituents alone or in combination could also be very relevant.
Point 4: L275 There is no reasoning or verification regarding the permanence of off-flavor or residues, thus limiting any discussion regarding toxicity and long time effects. A targeted GC-MS analysis of the grain at the end of the experimental period would be useful to verify concretely and quantitatively the actual presence of residues.
Response 4: Thank you for this remark. We are not able to make these analyses anymore but we can discuss about this lack and the importance of including GC-MS analysis in middle and long terms toxicity tests in the further studies as well as the importance of analyzing residues both in surfaces and inside the grain to allow a better explanation of influence of exposure duration. However, from a practical point of view, results of mortality were recorded after 7 days, if unsufficient effects are observed at that time, the oil is not suitable for insect control.
To consider your suggestion, we propose to add this text at the fourth § of the 4.4. duration of exposure:
Study must be carried out on the combined influence of evaporation and absorption by grains of essential oils in order to demonstrate its toxicity persistence in times. In further studies, it is a priority to include GC-MS analyses of treated wheat that allowed scientists to determine the behavior of essential oils and its remanence at the surface and inside the treated wheat until the end of experiment. This factor is essential to control insect pests that lay eggs into the grain, which causes a delay between treatment and the potential contact with the insecticide product by emerging individuals.
Point 5 : L150 it is not clear how the contribution of these compounds was determined in the absence of tests done with pure compounds. Also in this case, the use of an in-depth multivariate analysis could help, although some experiments on single pure compounds could also be very useful to minimize the risk of variable responses due to the use of often very changing mixtures such as essential oils. Such approach would also provide a new dimension to the manuscript, whose novelty is somehow limted.
Response 5: Thank you for your clever remark. In this study, we just observed that at the concentration tested these essential oils have not a strong effect on S. granarius and supposed that it would be the same for the pure major compounds at the concentration tested.
This work is a first step in a much larger study where we will eventually work on major compounds of most toxic essential oils in the goal to identify which compound(s) are responsible of their toxicity and to identify the presence of synergy between compounds. For instance, first results of second step experiment showed us that eugenol was fully responsible of the toxicity of Ocimum sanctum and Eugenia carryophyllus.
Unfortunately, for this first step study, we do not investigate major compounds of the 25 essential oils tested. It is true that antagonism/synergism could exists inside oils and that major compounds could be more or less toxic as a stand-alone. So, we propose to remove the name of major compounds in this part of the results and change it by essential oils tested. In addition, we have included in the conclusion of this article a perspective point where we mention the importance of carrying out study to understand in more detail the role of each constituent in the insecticidal effect.
Your remark on the variability of essential oils is also of the most importance. It is well known that the composition of essential oils can vary from one geographical area to another, from one season to another in the same place (for the same chemotype). We have added a sentence to this effect in the conclusion section of our publication to draw the reader's attention to the great importance of checking the precise composition of each essential oil:
“As essential oils are very variable composition product, study need to be achieved to identify clearly all compounds responsible of the insecticidal toxicity of these three essential oils to avoid variable response to the future treatment. More investigations need to be done on the mechanism of action of these oils, including the role of minor components, both on insects and mammals to secure their industrial use. “
Point 6: L190: Was toxicity to these insects observed with similar values? More or less intense?
Response 6: It is difficult to say in the sense that protocols are quite different, and the influence of grains is not take into account in these studies. For instance, Plata-Rueda et al. 2017 used topical application, Yang et al. fumigation in an unsealed box and Ho et al. used contact toxicity test with filter paper.
In our case, all paths of intoxication are potentially present (fumigation, contact, ingestion) but we do not know exactly at which level.
In addition, the fact that species are also different prevents us from making direct comparisons. For example, laboratory work on S. granarius and S. oryzae showed great differences of toxicity for a same essential oil (unpublished data).
The only study that could be compared here is the study you mentioned below : Zohry et al. 2020. Globally, they used the same protocol than us by treating wheat with essential oil and acetone mixture. If we consider that 10mL of wheat correspond to 8g (we checked it in our laboratory), they obtained 100% mortality with approximately 12µL of oil per 8g of wheat and we obtained the same with 20µL/8g. However, without precision on composition of the oil, we are not able to know if they treat grain with a similar product than us.
To answer to your last comments (points 6, 7 and 8) and include recent bibliography proposed below, we add this text in the discussion 4.1., § 3:
Essential oils toxicity of M. avensis [44], G. procumbens [45] and E. dives [46] as well as geraniol (main compound of T. vulgaris essential oil) [47] was also been highlighted for their activities against various stored product pests. In addition, Yazdgerdian et al. [45] identified G. procumbens as the most toxic oil, both by fumigation (6.8 µL/L air) and contact on treated wheat (0.235µL/g), among 5 essential oils tested on S. oryzae. These results confirm the toxicity of G. procumbens observed in our study. However, although many studies highlighted toxicities of essential oils, lack of a common protocol or of major compounds description often prevent from reliable and univocal comparison. For example, in the study of Teke et al. (2020) the fennel essential oils applied on S. granarius contains 71.64% of estragol, which closely resembles the composition of the basil oil in our study (73.43% estragol). However, in their case they realized topical application without grain presence which is quite different that in our case.
At the opposite, Zohry et al. (2020) tested toxicity of 10 essential oils on S. granarius by exposure to treated wheat in a protocol closed to this study. Garlic oil was identified as the most toxic one with a concentration of oil per grams of grain similar to ours. However, no precision on composition of EOs are available in their publication, which do not allow a deeper comparison. Further studies on the evaluation of the industrial potential of essential oils need to be based on a common protocol taking into account the influence of the media (Lee et al. 2004) and a full description of the composition of essential oils.
Point 7: L192 Some papers cited report toxicity expressed per mg / cm3. Would it be possible to extrapolate the data obtained in these experiments in the same way, to facilitate comparison?
Response 7: Comparison would stay quite difficult due to the presence of wheat. In another laboratory works (not published), we shown that wheat in presence of essential oils can adsorb gaz (by fumigation application) and/or liquids (by contact application). Quantification of this effect is not known for all essential oils discussed here and it should be quite difficult to compare results with other studies.
In addition, mg per volume is often use for fumigation studies. Here, insects are in contact with treated grains and could also eat the product which prevents direct comparisons. In addition, the fact that the Falcon tubes are not fully closed to not asphyxiate the insects in 24h and seven days experiments prevent us to define a volume.
The fact that very few studies use treated wheat to analyse EOs toxicity prevent us to compare our results in a deeper way.
Point 8 : It might be useful to integrate the discussion with some recently published articles dedicated to the same topic, also to highlight the heterogeneity of the studies that prevents reliable and univocal comparisons.
Response 8 : Please look at the answer to point 6.
Our modified manuscript is available. All corrections are visible in the text as desired by the editor.
We would like to thank you again for taking the time to correct our manuscript.
Best regards,
Sébastien Demeter
Reviewer 2 Report
The manuscript is well written with significant laboratory results on the impact of essential oils as insecticides on the granary weevil Sitophilus granarius.
In Material and methods, sufficient details of the techniques are provided; the significance of the Results and Discussion are well pointed out.
Please note that you'll have to move all the references after the one inserted.

Author Response
Dear reviewer,
I would like to thank you for the time you devoted to proofreading our manuscript.
The reference proposed has been added to the text and the numbers of other references has been modified.
We appreciated all the corrections you made to our manuscript and accepted all.
Our modified manuscript is available. All corrections are visible in the text as desired by the editor.
Best regards,
Sébastien Demeter
Reviewer 3 Report
Demeter et al. aimed to determine the insecticidal efficacy of a variety of essential oils (EO) against the granary weevil, Sitophilus granarius. The authors applied different concentrations of EO directly to wheat grains and tested the toxicity of 24 EO by measuring insect mortality after 24 hours and 7 days of exposure. Sixteen EO demonstrated efficacious killing, with garlic-derived EO showing the highest toxicity at the lowest doses at both time points. Most of the EO killed rapidly and, except for three EO, the insecticidal effect did not persist for 7 days. Furthermore, the authors sought to shed light on the plausibility of using the identified effective EO in real-world applications by including discussion of EO availability, cost, as well as their health implications. Under these conditions, garlic oil extracts were shown to be too expensive, had more limited availability, and had a higher mammal toxicity than some of the other EO. Contrary to this, oil extracts from Gaultheria procumbens, Mentha arvensis and Eucalyptus dives appeared to offer the best toxicity/cost value.
Overall, the science is sound and the findings are important for advancement of protection of stored food products against pests. However, there are some points that need clarification:
Clarification needed/general questions:
-Line 59: What do you mean by “bringing together all those toxicities”?
-You state that 9 EO did not perform well and moved on with 16 EO. 9+16=25 EO. There are 24 EO listed in Table 1; is there an additional EO not listed in Table 1?
-Why did you choose to use 7 days as the later time point? Is this related to hatching time such that at 7d, the EO should kill newly hatched neonates? Is there any information about whether these EOs kill eggs?
-Line 155: Do you mean “positive” relationship, since your data generally suggests the higher the concentration, the higher the mortality?
-If garlic EO kills with very low concentrations, very little would be needed. Is this included in your cost analysis?
-Line 188: Do you mean “lower” LD90 since it takes very little of the compound to kill the pest?
-If EO has low persistence, do you expect that multiple treatments are required to manage these pests (either from the hatching of pre-laid eggs or from new infestations)? How does the cost of multiple applications compare to a traditional insecticide?
In addition, there are several stylistic, grammatical, spelling, or formatting errors that need to be corrected. Some are pointed out here and in the attached PDF.
-Edit discussion for flow/ease of reading
-One-sentence paragraphs that should be combined with either the preceding or subsequent paragraphs.
-Table 2-difficult to tell which data/major compounds go together simply because of spacing used. Justify all to top of the cell rather than vertically centering. Also, re-label “control” column to read “control mortality (24h)”
-Table 4-top justify the EO column
-The use of a period verses a comma to indicate decimal places is inconsistent throughout manuscript
-Lines 101-104: this information is already presented in lines 80-83
-Some of the EO names are misspelled throughout the manuscript (Cuminum cyminum, Gaultheria procumbens, Vetiveria zizanioides).
-Provide a citation for the method of determining mortality (light shining in insect eyes)

Author Response
Dear reviewer,
I would like to thank you for the time you devoted to proofreading our manuscript.
I will answer your questions one by one :
Point 1: -Line 59: What do you mean by “bringing together all those toxicities”?
Response 1: We mean that by working on treated wheat authors take into account fumigation, contact and possibly ingestion toxicities. I changed the sentence by : “. A few less have worked on treated grains taking into account contact, fumigation and ingestion intoxication paths together [32].” Is this acceptable to you?
Point 2: You state that 9 EO did not perform well and moved on with 16 EO. 9+16=25 EO. There are 24 EO listed in Table 1; is there an additional EO not listed in Table 1?
Response 2: Thank you for this remark. It is a mistake Matricaria recutita has been removed from the table 1 by error. I put it back and change 24 in 25 in all the manuscript.
Point 3: Why did you choose to use 7 days as the later time point? Is this related to hatching time such that at 7d, the EO should kill newly hatched neonates? Is there any information about whether these EOs kill eggs?
Response 3: Initially, the duration of seven days was chosen because previous evaporation tests on some EOs shown us a quick evaporation of essential oils at 28°C meaning that the first 24h should be the peak of exposure. Also, seven days was chosen to take into account the fact that physiological impacts of certain oils could take few days to kill the insects.
Some essential oils are known to have a hatching inhibitory action on eggs but we didn’t investigate this here.
Point 4: Line 155: Do you mean “positive” relationship, since your data generally suggests the higher the concentration, the higher the mortality?
Response 4: Yes, sorry for this mistake, we changed it.
Point 5: If garlic EO kills with very low concentrations, very little would be needed. Is this included in your cost analysis?
Response 5: Yes, in comparison to M. arvensis for instance, A. sativum is 5 to 10 times more expensive but only 3 time more efficient in terms of quantity. In addition to his price, this oil contains toxic compound that can create a problem both in its production and its utilization. Those two factors indicate us that it could not represent the best choice for an “ecofriendly/healthy” insecticide.
Point 6: Line 188: Do you mean “lower” LD90 since it takes very little of the compound to kill the pest?
Response6: Yes, sorry for this mistake, we changed it.
Point 7: If EO has low persistence, do you expect that multiple treatments are required to manage these pests (either from the hatching of pre-laid eggs or from new infestations)? How does the cost of multiple applications compare to a traditional insecticide?
Response7: From discussions with stockholders, we think that multiple treatments is not a solution. First, price of a treatment with essential oil will be higher than actual pesticides even for a single treatment. Second, stockholders do not want to move their stock for a second treatment because of the extra cost and work that it generates. The solution should be a mixture of essential oils that allow to kill quickly the adult population (less than 24h) and that is also able to kill hidden forms of insects by fumigation for instance.
Concerning new infestations, without a strong work on formulation, essential oils could be use as curative solution but not as a preventive one.
Point 8: In addition, there are several stylistic, grammatical, spelling, or formatting errors that need to be corrected. Some are pointed out here and in the attached PDF.
Response 8: All corrections pointed out in the text were accepted.
Point 9: One-sentence paragraphs that should be combined with either the preceding or subsequent paragraphs.
Response 9: Thank you for this remark, we changed it.
Point 10: Table 2-difficult to tell which data/major compounds go together simply because of spacing used. Justify all to top of the cell rather than vertically centering. Also, re-label “control” column to read “control mortality (24h)”
Response 10: Thank you for this remark, we followed your suggestions.
Point 11: Table 4-top justify the EO column
Response 11: Thank you for this remark, we followed your suggestions.
Point 12: The use of a period verses a comma to indicate decimal places is inconsistent throughout manuscript
Response 12: We changed all commas by periods.
Point 13: Lines 101-104: this information is already presented in lines 80-83
Response 13: Thank you for highlighting this mistake.
Point 14: Some of the EO names are misspelled throughout the manuscript (Cuminum cyminum, Gaultheria procumbens, Vetiveria zizanioides).
Response 14: Thank you for this remark, we corrected it.
Point 15: Provide a citation for the method of determining mortality (light shining in insect eyes)
Response 15: We did not find another article that use light to determinate death of Sitophilus granarius. This technique is very efficient and has been developed following difficulties to identified knocked down from dead individuals. However, it is known that S. granarius is repelled by light (Gopal § Benny, 2018). This reference was added to the text.
I hope that the answers to your questions are satisfactory. Please do not hesitate to tell us if you need more details.
Our modified manuscript is available. All corrections are visible in the text as desired by the editor.
We would like to thank you again for taking the time to correct our manuscript.
Best regards,
Sébastien Demeter
Reviewer 4 Report
I have any doubt whether thyme oil came from Thymus vulgaris. Though there are many different chemotypes of Thymus vulgaris essential oil, but it contains thymol, carvacrol as the dominant components, and in smaller amounts: p-cymene, α- and β-pinene, α-terpinene, limonene, myrcene, linanool, linalool acetate, borneol, borneol acetate, α-terpinolene and 1,8-cineole. It can be assumed that investigated essential oil come from lemon thyme (Thymus × citriodorus L.) and contains mainly geraniol. Perhaps it will be possible to establish a commercial price for this oil [Omidbaigi R, Fattahi F, Alirezalu A. Essential Oil Content and Constituents of Thymus × citriodorus L. at Different Phenological Stages. Jeobp 2009; 12 (3): 333 – 337]; [Klaric MS, Kosalec I, Mastelic J at al. Antifungal activity of thyme (Thymus vulgaris L.) essential oil and thymol against moulds from damp dwellings. Lett Appl Microbiol 2007; 44:36-42.
Omidbeygi M, Barzegar M, Hamidi Z at al. Antifungal activity of thyme, summer Savory and clove essential oils against Aspergillus flavus in liquid medium and tomato paste. Food Control 2007; 18:1518-23.
Eteghad SS, Mirzaei H, Pour SF. Inhibitory effects of endemic Thymus vulgaris and Mentha piperita essential oils on Escherichia coli 0157:H7. Res J Biol Sci 2009; 4(3):340-4.
Sokovic MD, Vukojevic J, Marin PD et al. Chemical composition of essential oils of Thymus and Mentha species and their antifungal activities. Molecules 2009; 14:238-49.
Sartoratto A, Machado ALM, Delarmelina C et al. Composition and antimicrobial activity of essential oils from aromatic plants used in Brazil. Braz J Microbiol 2004; 36:275-80.
Imelouane B, Amhamdi H, Wathelet JP et al. Chemical composition and antimicrobial activity of essential oil of Thyme (Thymus vulgaris) from Easter Marocco. Int J Agric Biol 2009; 11(2):205-8.
Standen MD, Connellan PA, Leach DN. Natural killer cell activity and lymphocyte activation: Investigating the effects of a selection of essential oils and compounds in vitro. Int J Aromather 2006; 16:133-49.
Inouye S, Uchida K, Abe S. Vapor activity of 72 essential oils against a Trichophyton mentagrophytes. J Infect Chemother 2006; 12:210-6].
Please correct the Lamiaceae family name in introduction section.
Author Response
Dear reviewer,
I would like to thank you for the time you devoted to proofreading our manuscript.
I will answer your questions one by one :
Point 1: I have any doubt whether thyme oil came from Thymus vulgaris. Though there are many different chemotypes of Thymus vulgaris essential oil, but it contains thymol, carvacrol as the dominant components, and in smaller amounts: p-cymene, α- and β-pinene, α-terpinene, limonene, myrcene, linanool, linalool acetate, borneol, borneol acetate, α-terpinolene and 1,8-cineole. It can be assumed that investigated essential oil come from lemon thyme (Thymus × citriodorus L.) and contains mainly geraniol. Perhaps it will be possible to establish a commercial price for this oil
Response 1: Concerning a possible mistake on Thymus vulgaris CT geraniol, we transmitted your doubts to the R&D services of Pranarom industry (the supplier that sold us this essential oil).
They assure us that there is no error in the name of this oil and that it clearly distinguishes it from Thymus x citriodorus, lemon thyme. I hope this verification with the company removes the doubts you had.
Concerning its price, we didn’t find a producer price as for others oils. Indeed, we found price for 50-100kg (+/-1000 euros/kg) passing through some company but it cannot be compared to others. For the same company from which we identified this price, Mentha arvensis is 5 times more expensive that what we found at producer price for instance. We are sorry to not be able to provide a producer price for this essential oil.
Point 2 : Please correct the Lamiaceae family name in introduction section.
Response 2: Spelling of Lamiaceae in introduction section has been corrected.
Our modified manuscript is available. All corrections are visible in the text as desired by the editor.
We would like to thank you again for taking the time to correct our manuscript.
Best regards,
Sébastien Demeter
Round 2
Reviewer 1 Report
The authors have provided an adequate revision of their paper.